# Highly Strong Interface Adhesion of Polyester Fiber Rubber Composite via Fiber Surface Modification by Meta-Cresol/Formaldehyde Latex Dipping Emulsion

**DOI:** 10.3390/polym15041009

**Published:** 2023-02-17

**Authors:** Xiangze Meng, Le Kang, Xin Guo, Xiaohao Tang, Li Liu, Mei Shen

**Affiliations:** 1Engineering Research Center of High Performance Polymer and Molding Technology, Ministry of Education, Qingdao University of Science and Technology, Qingdao 266042, China; 2College of Polymer Science and Engineering, Qingdao University of Science and Technology, Qingdao 266042, China

**Keywords:** adhesive, composites, polyester fibers, rubber, m-cresol formaldehyde resin

## Abstract

As a skeleton material, polyester (PET) fiber can significantly improve the strength and durability of rubber composites, but the interfacial adhesion between polyester fiber and rubber is poor due to the chemical inertia of PET fiber surface. Resorcinol-formaldehyde-latex (RFL) impregnating solution is usually used to treat PET fibers, but RFL contains toxic components such as resorcinol, which is harmful to the human body. A simple and less toxic resin-impregnating system cresol-formaldehyde-latex (CFL) was obtained by alternating resorcinol with low-toxicity cresol and m-cresol formaldehyde resin was synthesized from m-cresol and formaldehyde. CFL (m-cresol formaldehyde resin latex) systems with different C/F mole ratios and CF resin/latex ratios were adopted to modify the surface of PET fibers. The strip peeling adhesive and the H pull-out test results indicated that the PET fiber/rubber adhesion strength increased with the increase in the formaldehyde dosage and the CF resin content, and the peeling force value and the H-pull-out force of treated PET/rubber composites reached 7.3 N/piece and 56.8 N, respectively. The optimal choice of CFL adhesive system was obtained, when the C/F mole ratio was 1/2 and the CF resin/latex weight ratio was 0.23. This environment-friendly CFL dipping emulsion can be used as a new surface modification strategy as it can remarkably enhance the interfacial adhesion of PET/rubber composites.

## 1. Introduction

Rubber products are widely used in all aspects of our life and industry, such as tires, conveyor belts, and V-belts [1,2]. In order to improve the strength and dimensional stability of rubber, fiber skeleton materials are usually added in the rubber products for reinforcement. The interface adhesion between fiber and rubber plays a vital role in the overall performance of rubber products. Polyester fiber is the most used synthetic fiber with excellent mechanical properties and thermodynamic stability [3]. There are only the hydroxyl groups at both ends of the molecular chains, the PET fiber does not contain any other functional groups on the chain. Therefore, the PET fiber is a hydrophobic fiber. Although the ester groups present certain activity, the reactivity of the benzene ring and the methylene group is low and the overall PET fibers are inert. Even worse, the rubber commonly used is non-polar in the field of tires, conveyor belts, and V-belts. The difference in polarity leads to a weak interaction between the two kinds of materials. It is difficult for polyester fiber to form a good interface layer with the rubber matrix, which limits its application in the rubber industry. Therefore, it is necessary to modify the fiber surface to improve the interface interaction between rubber and fiber. 

At present, RFL resin-impregnating system is traditional and mature, of which resorcinol-formaldehyde (RF) resin forms a three-dimensional network structure to provide fatigue resistance and polarity, and vinyl-pyridine latex inserted into RF, improve the adhesion between rubber and fiber as a cross-linker with rubber [4,5]. However, the RFL system is toxic and hazardous to personal health, such as respiratory irritation symptoms and skin damage for long-term exposure to low concentrations of resorcinol. In addition, serious poisoning or death is going to occur because of exposure to a high concentration of resorcinol for a long time [6]. In order to find an environmentally friendly way to improve the adhesion between fiber and rubber, various process methods have been carried out. Fiber surface modification can generally be divided into two categories: physical modification and chemical modification. Physical modification method mainly needs external energy to change the properties of the fiber or the properties of the fiber surface groups, or require additional substances to cover the surface defects of the polyester fiber, such as physical surface etching and graft modification to modify the polyester fiber. Chemical modification is to use certain chemical reagents to chemically react with the molecular chains of the polyester fiber surface in order to introduce active groups on the surface of the fibers, or to carry out secondary grafting reactions after the introduction of active groups, thereby increasing the polarity of polyester fibers. Such as, electron beam (EB) irradiation is an ideal modification method since the treatment is energy saving and environmentally friendly, which has no pollution to the environment [7]. However, the intensity of high energy ray irradiation is the key to the modification effect. If the intensity is too high, great damage to the fiber body structure is caused inevitably to reduce the fiber strength and affect subsequent processing. Enzymatic process for activation has shown great advantages as a biological enzyme technology because of its mild action conditions, environmental friendliness, and almost no damage to fibers [8]. However, the current enzymatic surface modification of polyester fibers has too much problems to be resolved, such as poor modification effect and low reaction rate. Plasma polymerization is a kind of method to increase the active groups on the surface of polyester fibers to improve the adhesion performance by the aid of expensive equipment [9]. UV-assisted surface modification needs ultraviolet radiation technology, one of the high energy ray irradiation modification methods. With its strong applicability, environmental protection, high efficiency, and safety in the use process, ultraviolet radiation technology has been comprehensively used in the functional modification of various textiles [10]. However, polyester fiber has poor antiultraviolet performance to get rid of damaging from ultraviolet radiation. Microwave modification activates the fiber surface by etching the fiber surface. The disadvantage of microwave treatment is that the strength and modulus of the fiber is reduced and the specific surface area of the fiber is increased because of the separation and refinement of the fiber after being modified [11,12]. Grafting mercapto hyperbranched polysiloxane modification and coupling agent treatment can effectively improve the surface properties of polyester fibers, enhance the reactivity, improve the wettability of the fiber surface, and effectively increase the oxygen content [13]. However, the modification method of coupling agent treatment is more complicated in processing and strict in processing conditions, leading to less promotion in practical applications. Coating method is to coat a layer of polymer resin or low molecular substances on the surface of the polyester fiber. The disadvantage of coating treatment is that the type of coating selected is limited by the requirements of the coating agent in good compatibility with the surface of the fibers and same polar groups or active center. The application range of the obtained composite is limited because the thickness of the coating is difficult to control to obtain good treatment effect for poor process operability and repeatability [14,15,16]. Oxidative acid etching is the most basic, simple, and effective chemical modification method. Although the surface roughness and the number of active groups of polyester fibers increase, the greater the roughness, the greater the number of active groups with destroyed crystallization and the more serious the damage to the fiber structure itself [17]. Laser modification can achieve the purpose of fiber surface activation, but instruments and equipment used in the process are complex in the requirements and high in the cost [18]. Chemical grafting method can protect the polyester fiber structure to maintain the original fiber strength, but also it can increase the interfacial cohesion of the aramid fiber surface. However, the content and composition of functional group elements change significantly during the modification process, and only a chemical reagent featuring grafting reaction with benzene rings or hydroxyl groups can be applied in this method [19,20,21,22,23,24]. 

However, physical methods such as laser and plasma processing not only require special instruments and specified conditions, but also they usually cause damage to the fibers, resulting in higher costs and a decrease in fiber strength. In addition, traditional surface treatment methods, such as etching and grafting, damage the fiber as well. Hence, it is necessary to find a simple, reliable, and less toxic method for fiber surface modification without sacrificing fiber strength and bonding properties.

In this study, a simple and less toxic resin-impregnating system cresol-formaldehyde-latex (CFL) was obtained by alternating resorcinol with low-toxicity cresol and m-cresol formaldehyde resin was synthesized from m-cresol and formaldehyde with the least toxicity. PET fiber/SBR composites were prepared. The optimum treatment conditions of C/F and CF resin/L were studied, and the synthesized CF resin was characterized. The strip peel adhesive and the H pull-out tests were carried out by universal testing machine. It is demonstrated that the highly strong interface adhesion between rubber and PET fabrics can be obtained via fiber surface modification by meta-Cresol/formaldehyde latex dipping emulsion to replace the traditional RFL impregnation emulsion.

## 2. Materials and Methods

### 2.1. Materials

Vinyl-pyridine latex (40%) was purchased from Gaoming Chemical Co., Ltd., Shanghai, China. Meta-Cresol (C), Formaldehyde (F) and NaOH were purchased from Sinopharm Chemical Reagent Co., Ltd., Shanghai, China. Polyester fabrics were obtained from Taiji Industry New Materials Co., Ltd., Yangzhou, China. Polyester fabrics treated by RFL were obtained from Qingdao Tianbang Wire Industry Co., Ltd., Qingdao, Shandong, China. Styrene Butadiene Rubber (SBR1502E) was purchased from Sinopec Qilu Petrochemical Co., Ltd., Zibo, China. All the rubber ingredients were industrial grade and used as received.

### 2.2. Synthesis of M-Cresol/Formaldehyde Resin and Preparation of Different CFL Emulsions

#### 2.2.1. Synthesis of M-Cresol Resin and Preparation of CFL Emulsions with Different M-Cresol/Formaldehyde Molar Ratios

The m-cresol/formaldehyde resin was prepared in various C/F molar ratios (C/F = 1/1.2, 1/1.4, 1/1.6, 1/1.8, 1/2.0, 1/2.2, 1/2.4, 1/2.6) following the process of Figure 1. Under the conditions of alkali catalysis, the addition reaction of formaldehyde is dominant and the condensation reaction is slow. The resin initially formed is resole m-cresol formaldehyde resin. It is mainly divided into two reaction stages: (1) m-cresol and formaldehyde first undergo addition reaction to generate 1~3 hydroxymethyl m-cresol; (2) hydroxymethyl m-cresol is further condensed into initial resin or thermosetting resin, and then further cross-linking can be carried out under high temperature conditions. Firstly, C/F in the ratio of 1/1.2 was added to deionized water (500 mL), then the emulsion was mildly at 80 °C for 60 min after adjusting the PH value of the emulsion to 8 by 1% NaOH emulsion. Subsequently, 785 g of the VP latex was added to the above emulsion to obtain a uniform emulsion by stirring. The C/F-VP latex was noted as CFL. CFL emulsions with different m-cresol/ formaldehyde molar ratios were prepared and marked as 1/1.2, 1/1.4, 1/1.6, 1/1.8, 1/2.0, 1/2.2, 1/2.4, 1/2.6, accordingly. 

#### 2.2.2. Preparation of CFL Emulsions with Different Resin/Latex Wet Weight Ratios

The CFL dipping emulsion was prepared with C/F molar ratio (C/F = 1:2) and different CF resin contents (wet weight ratio: 11%, 13%, 15%, 19%, 21%, 23%, 25%, 27%, and 29%, respectively) after being sealed and cured for 24 h. The preparation process of different CFL emulsions and the detailed chemical interaction between PET/CFL and rubber were shown in Figure 2.

### 2.3. Preparation of PET/CFL Composites

#### 2.3.1. Impregnation of PET Fibers in CFL Emulsions

The dipped PET fibers were obtained by immersing fibers into different CFL emulsions and then drying at 180 °C for 2 min and curing at 200 °C for 5 min. The preparation process of different CFL emulsions and the fiber impregnation process are shown in Figure 2.

#### 2.3.2. Preparation of PET/Rubber Composites

To prepare the H pull-out and strip peeling adhesion test samples, a standard rubber formula is used as followings: SBR1502 100, stearic acid 1.5, zinc oxide 2, carbon black N330 40, N-cyclohexylbenzothiazole−2-sulphenamide (CBS) 1.5, sulfur 1.5, 9,9-dimethylcarbazine (BLE) 2, in phr (per hundred rubber). According to the ASTM D4776−1998, two ends of one PET fibers treated by CFL was buried into two paralleled rubber bulks, as shown in Figure 3a, with a pretension provided by a 50 g weight. The composite was then trimmed into H-shapes for testing as Figure 3a after being vulcanized at 160 °C for 14 minutes under 15 MPa. The optimum cure time t_90_ is determined by RPA tester at 160 °C for 30 min. In addition, vulcanisation time of sample is two minutes more than t_90_ from the test results, considering the sample thickness.

One end of one long enough dipped fiber was knotted and fixed in the wire groove at one end of the mold, then the fiber was reciprocally arranged between the wire grooves at both ends of the mold so that each sample was arranged with 7 pieces, at last tension was applied by a weight of 50 g at another end of the fiber. The samples were cut to 25 mm wide reference to GB/T 40725−2021, after removing one fiber on each side of each sample as in Figure 3b.

### 2.4. Test Instruments and Methods

The Fourier-transform infrared spectroscopy (VERTEX70 Fourier transform infrared spectrometer, North Billerica, MA, USA) was used to characterize the chemical structures of the CF resin via the TR-FTIR mode, from 500 cm^−1^ to 4000 cm^−1^ with a resolution of 4 cm^−1^ for 32 scans. The thermal stability of CF resins was tested by thermogravimetric analysis (METTLER-TOLEDO, Zurich, Switzerland). ^1^H-NMR of CF resin was measured by a Bruker AV-III 500 NMR spectrometer instrument at room temperature, with CDCL_3_ as a solvent and TMS as the internal standard peak. The morphology and interfacial elements changes were observed though a scanning electron microscope (A JEOL JSM−6700 F, Kyoto, Japan) equipped with an energy dispersive spectrometer (EDS) detector. The breaking strength of PET fibers was measured by single-axis tensile machine (GT-TCS−200, Gotech Testing Machines Inc., Taiwan, China) at the speed of 100 mm/min according to GB/T 30311−2013. A tensile tester was used to carry out the H pull-out tests at the speed of 100 mm/min according to ASTM D4776−1998 and the strip peeling adhesion tests at the speed of 100 mm/min according to GB/T 40725−2021.

## 3. Results and Discussion

### 3.1. Chemical Composition of the CF Resin

Figure 4 shows infrared spectra in the frequency region 500–4000 cm^−1^ to characterize the surface chemical structures of C/F resin with different molar ratios. The absorption peaks around 3300 cm^−1^ are assigned to stretching vibrations of phenolic hydroxyl and hydroxyl groups. The peaks at 1595 cm^−1^ and 1465 cm^−1^ are assigned to extension vibration of the double bond (C=C) on the benzene ring. With the number of formaldehyde groups increases, the hydroxymethyl groups bonded to the benzene ring increase, and the electronegativity difference between the two ends of the C=C bond on the benzene ring skeleton decreases, resulting in lower absorption peak intensity. The peaks around 1280 cm^−1^ are corresponded to stretching vibration of the C-O bond. The absorption peak at 1161 cm^−1^ is stretching vibration absorption peak of the ether bond (CH_2_-O-CH_2_). The peaks at 1020 cm^−1^ are corresponded to stretching vibration absorption of the methylol, the intensity of which increases as the number of formaldehydes increases. The peaks at 770 cm^−1^ and 690 cm^−1^ are the C-H bending vibrations of monosubstituted benzene, and the intensity of which decreases with the increase in hydroxymethyl groups on the benzene ring.

Figure 5 shows the TGA tests in a flowing N_2_ atmosphere to characterize the thermal decomposition process of the CF resin and analyze the changes of the chemical structures from 50 °C to 800 °C. The CF resin shows a weight loss of 5−15% in the range of 310–410 °C, which can be related to the decomposition of terminal hydroxymethyl groups of CF resin. With the number of formaldehyde groups increases, the hydroxymethyl groups increase, and the temperature of 5% weight loss decrease. This is consistent with the conclusion drawn in the work of Liu [25]. The second weight loss stage is at 410–610 °C, indicating the pyrolysis and cracking of methylene, and the dehydration of the phenolic hydroxyl group and the cyclization into carbon [26]. It can be seen the heat resistance of CF resin decreases as the ratio of C/F increases.

As shown in Figure 6, the multiple peaks at 7.17–7.05 ppm are corresponded to phenolic hydroxyl proton peaks. The multiple peaks at 6.61–6.74 ppm are assigned to aromatic ring hydrogen proton peaks. The peaks at 4.55–4.95 ppm are assigned to hydroxymethyl hydrogen proton, and 3.51–3.39 ppm are methylene bridge hydrogen proton peak. The peaks at 2.26–2.31 ppm are the proton peaks on methyl, which conforms to the characteristic structure of CF resin.

### 3.2. Adhesion and Strength Results

The H pull-out test is an evaluation method to determine the interfacial adhesion of fiber/rubber composites under shearing force. The strip peeling adhesion and the H pull-out test values of PET/CF fiber with different C/F molar ratios and CF resin contents are given in Figure 7. As seen in Figure 7a,b. The strip peeling force and the H pull-out force values of the fiber/rubber composites increase obviously with the formaldehyde dosage increasing. When the molar ratio of C/F is 1/2.0, the peeling force value reaches the maximum value of 7.4 N/piece and the H pull-out force reaches the maximum value of 53.9 N. The peeling of fiber/rubber composites, in which the fiber is not treated by any emulsion, is only about 1.0 N. Compared with the untreated fiber, the polyester fiber treated by CFL impregnation emulsion has an increase of about 86.48% in the peeling force value from rubber, due to the chemical bond and mechanical interlocking that are formed between the treated PET fiber and the rubber. The peeling force value increases with the molar ratio and reaches the maximum when the molar ratio is 1/2, because m-cresol and formaldehyde form CF resin, with better chain flexibility. However, the by-products of the product, and the cross-linked network density increases as the number of formaldehyde continues to increase, resulting in an increase in modulus and a decrease in the bonding effect.

As it can be seen in Figure 7c,d, the peeling force value reaches the maximum value of 7.3 N/piece and the H pull-out force reaches the maximum value of 56.8 N when CF resin/latex weight ratio was 0.23. The reason is that the increase in molecular weight and crosslinking density of resin makes a greater crosslinking contribution to the reinforcement of composite materials with the increase in resin content. However, when the excessive resin cures itself, the interface becomes brittle, with weak adhesiveness.

### 3.3. The Breaking Strength of PET

In order to demonstrate the influence on the breaking strength of PET fiber by the dipping process, the breaking strength of PET fiber before and after dipping is tested and shown in Figure 8. The results show that the breaking strength of the PET fiber decreased by less than 5%, which does not have any unfavorable effect on practical application. Due to the reaction of CF resin with the hydroxyl in the PET during the activation process, the hydrogen bonding forms between the molecular chains of the fiber and CF resin without breaking the PET molecular chain, which indicates that the CFL dipping emulsion will not cause damage to the mechanical properties of the PET fiber. It can be concluded that the dipping treatment of PET by CFL does not damage its breaking strength, thus guaranteeing the safety and mechanical properties of CFL treated PET/rubber composites.

### 3.4. Morphology of Composites

Figure 9 is the SEM image of the fiber surface after peeling off the PET/SBR composites treated with CFL dipping emulsion. From Figure 9a,b, there is little rubber existing on the surface of PET fiber after dipping treatment by the emulsion with C/F molar ratio of 1/1.2 and 1/1.6, indicating in poor interfacial adhesion properties for poor interactions with the rubber matrix. With less amount of formaldehyde used the adhesion failure occurs between the dipping layer and PET fiber for insufficient interfacial adhesion strength. As shown in Figure 9c, a thin layer of adhesive is residual on the PET fiber surface with dipping treatment by the emulsion with C/F molar ratio of 1/2. Owing to the dipping treatment by the C/F-latex and the reactions between the PET fiber and rubber, PET fiber treated by emulsion with C/F molar ratio of 1/2 presents enhanced surface properties compared to the PET fiber. It is indicated that the dipping layer has played an important role as the middle layer, to bond the fiber and rubber together. As shown in Figure 9d, much more rubber remains on the surface of the PET fiber surface treated by with C/F molar ratio of 1/2.6, which shows a strong interface adhesion between the treated PET fiber and rubber and proves that the failure location is in the rubber matrix with less modulus compared to PET fiber. The SEM results are well consistent with the previous H pull-out test and the strip peeling test result.

Figure 10 shows the changes of interface element ratio of PET fiber/rubber composites dipped with CFL (C/F = 1:2) emulsion. It can be seen that the sulfur element in the dip layer shows an obvious aggregation in the PET fiber/rubber interface region, but no sulfur-containing compound was added to the CFL dipping emulsion, due to the difference in solubility of sulfur between the rubber and the dipped layer. On the one hand, the aggregation of sulfur in the interface between PET fiber and rubber increases the crosslink density of the dip layer and enhances the modulus of the interfacial transition layer, benefiting the transfer of stress. On the other hand, it also provides new method for excellent PET fiber/rubber interface.

## 4. Conclusions

In this work, C/F resin was synthesized instead of high toxicity resorcinol formaldehyde resin to prepare impregnation emulsion, aiming to improve the interface interaction between PET fibers and rubber matrix, CFL dipping emulsion with different C/F mole ratios and CF resin/latex ratios were adopted to modify the surface of PET fibers. FTIR, NMR, and TGA were carried out to confirm the chemical structure of CF resin as expected, and the strip peeling adhesive and the H pull-out tests were performed to evaluate the interfacial adhesion strength of CFL treated PET fiber/rubber composites. PET fiber/rubber adhesion increases with the increase in the formaldehyde dosage and CF resin content. The peeling force value and the H pull-out force of treated PET fiber/rubber composites reached 7.3 N/piece and 56.8 N, respectively, which could be ascribed to the use of CF resin in the activation process, reacting with the hydroxyl in the PET and forming the hydrogen bonding between the molecular chains of the fiber and CF resin. At the same time, the interface interaction is further improved by the generation of mechanical interlocking effect between latex and rubber. The fracture of peeling tests occurs in the layer of rubber matrix. The optimal ratio of CFL adhesive system is obtained, and the C/F mole ratios combined with the CF resin/latex weight ratio are 1/2 and 0.23, respectively. The surface modification strategy proposed in this study can remarkably enhance the interfacial adhesion of PET fiber/rubber composites with great potential in practical application.

As polyester fiber is widely used in industry and life, the modification of polyester fiber is the inevitable way for its application and industrialization. The existing physical and chemical modification methods are becoming more and more mature. More and more physical and chemical intersection methods will also emerge as the times requires, aiming to obtain polyester composite materials suitable for different conditions through an environmentally friendly, safe, and efficient process, and more research and development of modified polyester fibers will be applied in industry. Finding environmental protection, safe, economical, efficient, and convenient modification methods is still the top priority in the application technology of polyester fiber modification research.

## Figures and Tables

**Figure 1 polymers-15-01009-f001:**
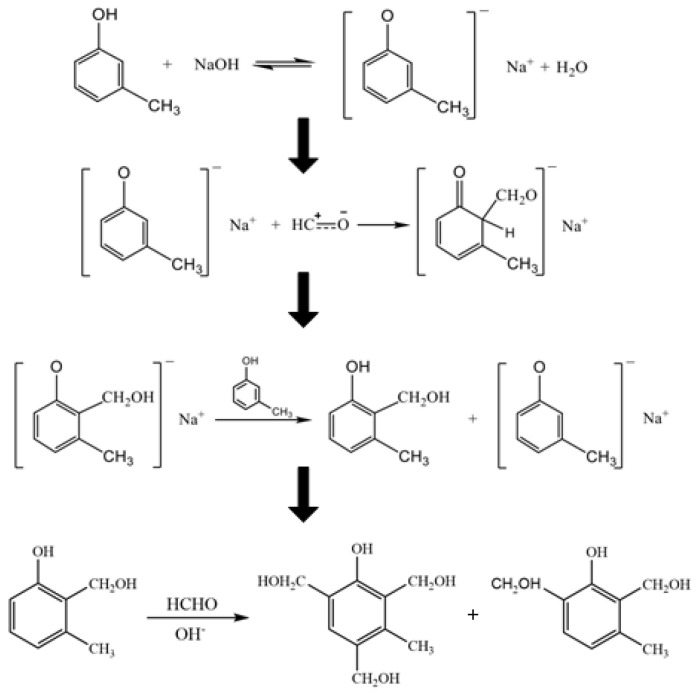
The synthesis scheme of m-cresol/formaldehyde resin.

**Figure 2 polymers-15-01009-f002:**
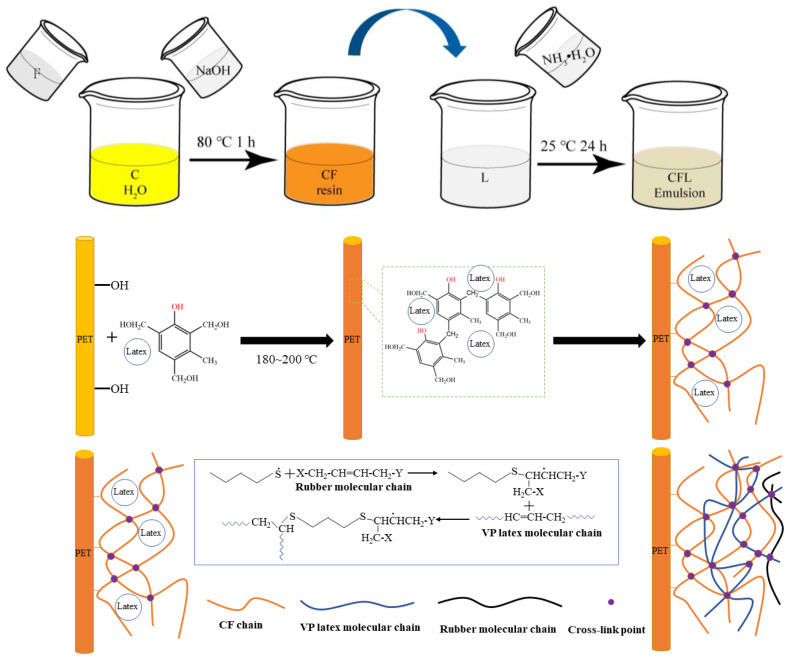
The preparation process of different CFL emulsions and the fiber impregnation.

**Figure 3 polymers-15-01009-f003:**
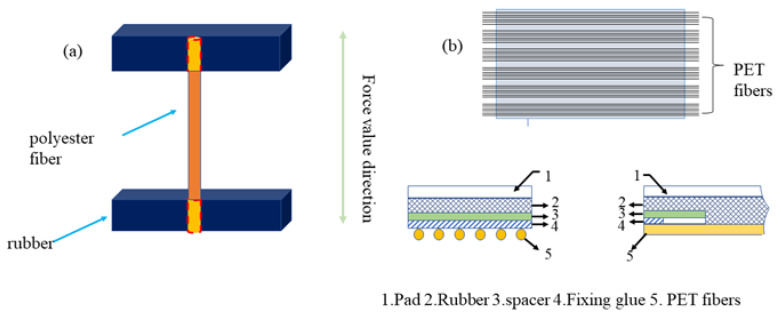
Schematic diagram of test specimens (**a**) H pull-out sample (**b**) strip peeling adhesion test sample.

**Figure 4 polymers-15-01009-f004:**
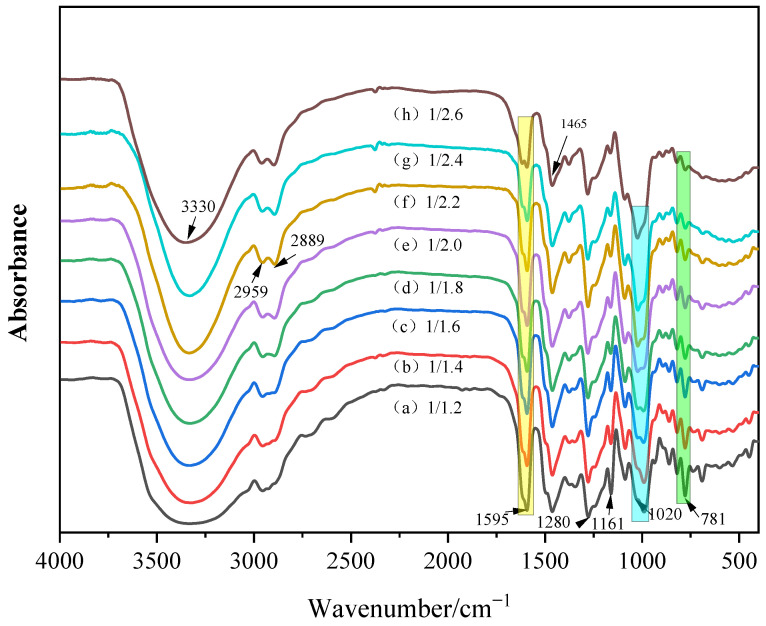
TR−FTIR spectra of CF resins.

**Figure 5 polymers-15-01009-f005:**
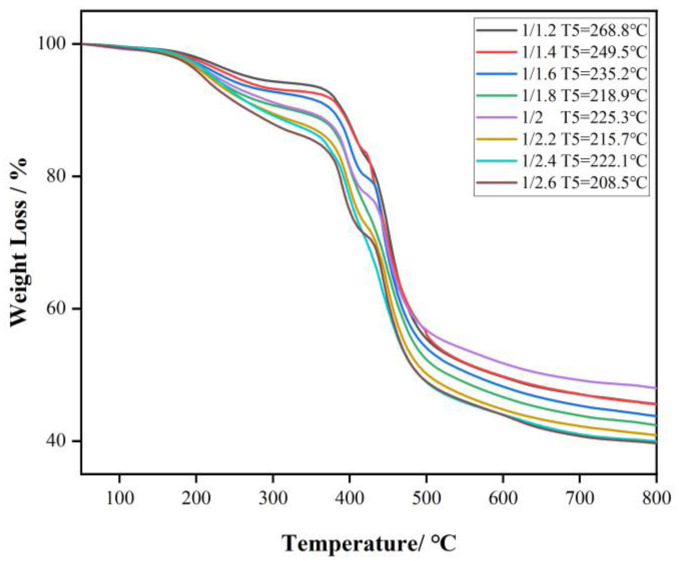
TGA thermograms of CF resins.

**Figure 6 polymers-15-01009-f006:**
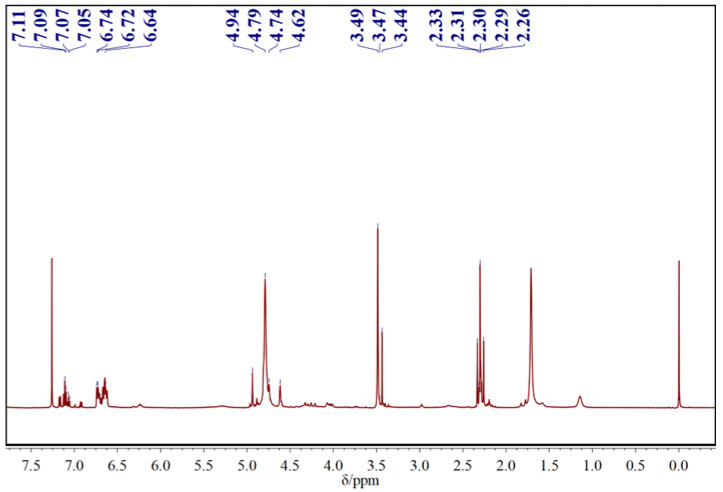
The ^1^HNMR spectra of m-cresol formaldehyde resin with a molar ratio of 1/2.0.

**Figure 7 polymers-15-01009-f007:**
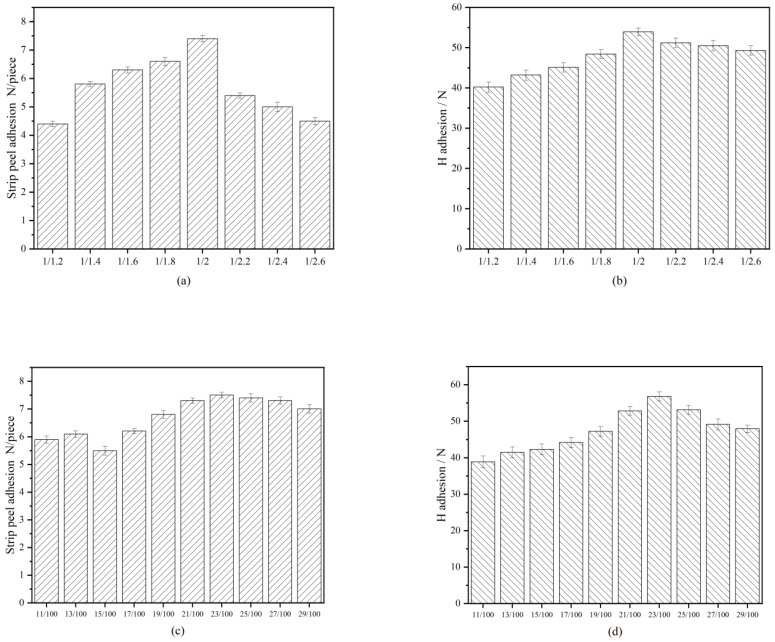
Adhesion strength (**a**) strip peeling adhesion and (**b**) H pull-out test of different m-cresol/formaldehyde molar ratios values. (**c**) The strip peeling adhesion and (**d**) H pull-out test of different wet weight ratios of m-cresol formaldehyde resin/latex dipping emulsion.

**Figure 8 polymers-15-01009-f008:**
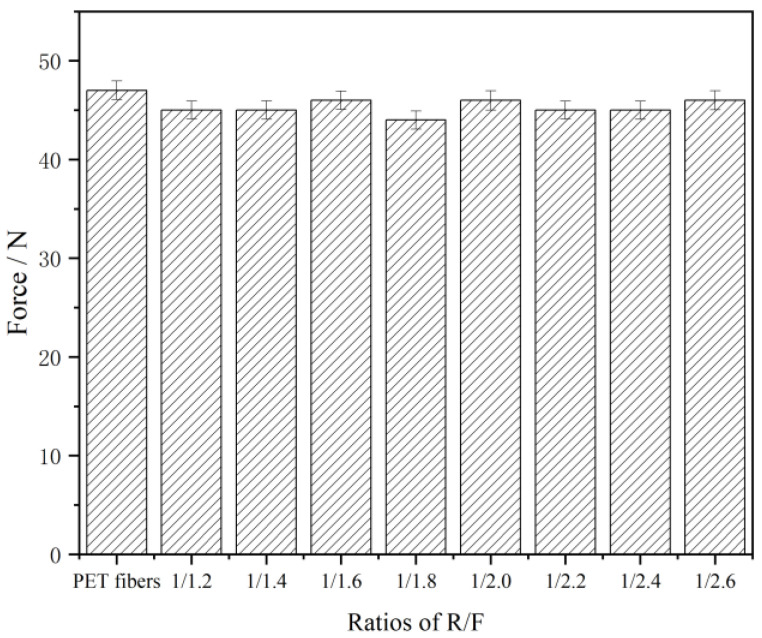
The breaking strength of PET fiber.

**Figure 9 polymers-15-01009-f009:**
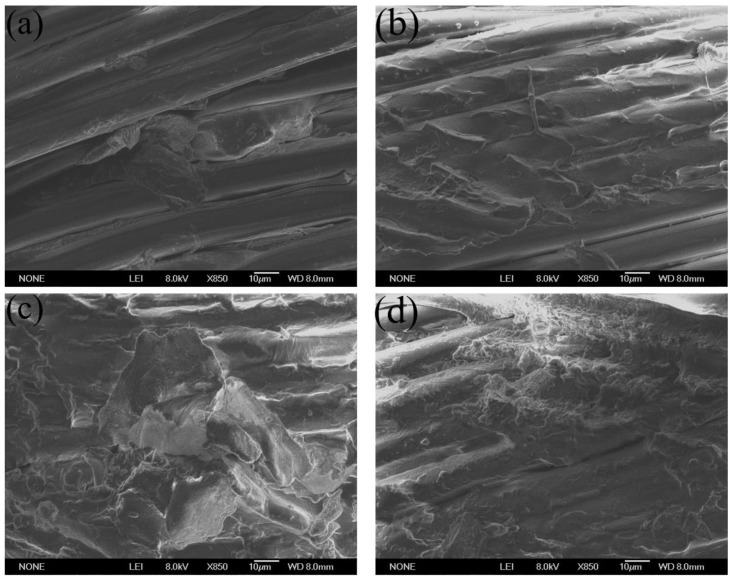
SEM images of PET/SBR composites impregnated with CFL emulsion. (**a**) C/F = 1/1.2, (**b**) C/F = 1/1.6, (**c**) C/F = 2.0, (**d**) C/F = 1/2.6.

**Figure 10 polymers-15-01009-f010:**
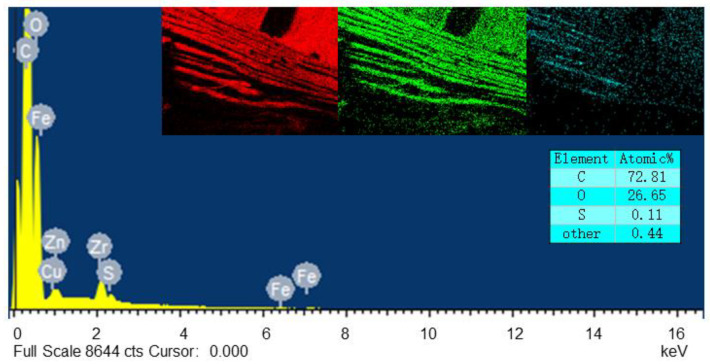
EDS Analysis of Element Content of Impregnated Layer on PET Fiber Surface.

## Data Availability

Not applicable.

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
