# Peer review of "Highly Strong Interface Adhesion of Polyester Fiber Rubber Composite via Fiber Surface Modification by Meta-Cresol/Formaldehyde Latex Dipping Emulsion"

_polymers, 2023, doi:10.3390/polym15041009_

Round 1
Reviewer 1 Report
The research article entitles ‘Highly strong interface adhesion of Polyester fiber rubber com- 2 posite via fiber surface modification by meta-Cresol/formalde- 3 hyde latex dipping emulsion” is found novel and interesting. The This study is a simple and less toxic resin-impregnating system cresol- 15 formaldehyde-latex (CFL) was obtained by alternating resorcinol with low-toxicity cresol and m- 16 cresol formaldehyde resin was synthe-sized from m-cresol and formaldehyde. This research will be more complete if some of these points are added. Hence this work can be accepted after a major revision with following comments.
1. Title is ok no changes required
2. Abstract is fund good and precise
3. Citation of style found wrong in the running text kindly correct it.
4. Few spelling mistakes found hence Spell check is recommended
5. What are the unanswered questions you found from literature?
6. What happens at the second weight loss stage in TGA analysis?
7. Why the density of cross-linked networks increases when the amount of 186 formaldehyde continues to increase?
Generally, this work is good enough for publication. And need above major revisions before acceptance. I convey my best wishes to authors.
Author Response
- Title is ok no changes required
No change.
- Abstract is fund good and precise
No change.
- Citation of style found wrong in the running text kindly correct it.
Incorrect style references have been corrected.
- Few spelling mistakes found hence Spell check is recommended
Spell check is done.
- What are the unanswered questions you found from literature?
Although physical treatment of fibers can improve the bonding strength between the fibers and the rubber matrix, it damages the fibers themselves in the strength. The new chemical methods with low toxicity have been used to treat the fiber surface in recent years, resulting in poor bonding strength between the fiber and the rubber matrix to compare with the traditional RFL impregnation system. Therefore, it is necessary to find an impregnation method with low toxicity and maintain fiber strength, and high-efficiency bonding.
- What happens at the second weight loss stage in TGA analysis?
The second stage of thermal weight loss mainly presents the methylene cleavage in the m-cresol formaldehyde resin and the dehydration reaction between the phenolic hydroxyl groups, which has been supported by the citation in this part.
- Why the density of cross-linked networks increases when the amount of 186 formaldehyde continues to increase?
The reaction functionality of m-cresol is three. When the amount of formaldehyde increases, the molar ratio of formaldehyde to m-cresol increases. As a result, more trimethylol m-cresol can be formed in the initial reaction, and the greater the amount of formaldehyde, the more trimethylol m-cresol is formed. Hence, the cross-linking density of m-cresol formaldehyde resin increases in the next process of heat curing.

Reviewer 2 Report
Dear Authors,
The manuscript entitled “Highly strong interface adhesion of Polyester fiber rubber composite via fiber surface modification by me-ta-Cresol/formaldehyde latex dipping
emulsion” has been reviewed.
The manuscript is very interesting and helpful and I found there only little things to correct, repair or explain.
Detailed comments are as follows:
1) Page 4, line 106: “N330” – better “carbon black N330” (not all scientist know, how carbon black are marked).
2) Page 4, line 107: “(CZ)” – the right and used abbreviation for N-cyclohexylbenzothiazole-2-sulphenamide is “(CBS)”.
3) Page 4, line 108: all the rubber chemicals are “in weight” – right should be “in phr (per hundred rubber)”.
4) Page 4, line 111: the vulcanization was at 160 °C for 14 minutes – how you set this time? Did you measure vulcanisation curve, or how you choose this time? Please explain it in text.
5) Page 7, figure 7: Figure 7b and 7d are same. Please add the right figure.
6) Page 8, figure 8: There are no error bars. I think, they should be by all figures.
7) Page 10, line 257: … “ration” …, should be “ratio”.
Author Response
- Page 4, line 106: “N330” – better “carbon black N330” (not all scientist know, how carbon black are marked).
Modified.
- Page 4, line 107: “(CZ)” – the right and used abbreviation for N-cyclohexylbenzothiazole-2-sulphenamide is “(CBS)”.
Modified.
- Page 4, line 108: all the rubber chemicals are “in weight” – right should be “in phr (per hundred rubber)”.
Modified.
- Page 4, line 111: the vulcanization was at 160 °C for 14 minutes – how you set this time? Did you measure vulcanisation curve, or how you choose this time? Please explain it in text.
The optimum cure time t90 is determined by RPA tester at 160 °C for 30 minutes. And vulcanisation time of sample is two minutes more than t90 from the test results, considering the sample thickness. Explanation is added in the text.
- Page 7, figure 7: Figure 7b and 7d are same. Please add the right figure.
Modified. Right figure is added.
- Page 8, figure 8: There are no error bars. I think, they should be by all figures.
Error bars are added in Figure 8.
- Page 10, line 257: … “ration” …, should be “ratio”.
Modified.
